# Examining community perspectives on integrated service delivery for tuberculosis, mental health and substance use disorder in Nigeria: A qualitative study

Martin Njoku[1], Charles Nwafor[1], Chinwe Eze[1], Okechukwu Ezeakile[1], Anthony Meka[1], Ngozi Ekeke[1], Iyama Francis[1], Daniel Egbule[1], Joseph Chukwu[2], Charles Esekhaigbe[3], Chijioke Osakwe[3], Edmund Ndudi Ossai[4]*, Chibuike Agu[4], Grace Bernard-Asadu[5], Chukwuma Anyaike[6], Clement Adesigbin[6], Obioma Chijioke-Akaniro[6], Tunde Ojo[7], Daniel C. Oshi[8], Beatrice Kirubi[9], Jacob Creswell[9], Ngozi Murphy-Okpala[1]

**1** RedAid Nigeria, Enugu, Nigeria, **2** German Leprosy and TB Relief Association, Enugu, Nigeria, **3** Initiative for Prevention and Control of Diseases, Nigeria, **4** Department of Community Medicine, College of Health Sciences, Ebonyi State University Abakaliki, Nigeria, **5** Department of Community Medicine, University of Nigeria Teaching Hospital Ituku Ozalla, Enugu, Nigeria, **6** National Tuberculosis, Leprosy and Buruli Ulcer Control Program Abuja, Nigeria, **7** National Mental Health Program, Department of Public Health, Federal Ministry of Health, Abuja, Nigeria, **8** Department of Community Health & Psychiatry, The University of the West Indies, Jamaica, **9** STOP TB Partnership, Geneva, Switzerland

* ossai_2@yahoo.co.uk

## Abstract

### Background

The concept of integrated service delivery was the focus for the envisaged essential health care under one roof. Despite being the central focus of essential health under one roof and a crucial principle of primary health care, relatively little is known about community perceptions on integrated service delivery in low-and middle income countries. This study was designed to examine community perspectives on integrated service delivery for tuberculosis (TB), mental health (MH) and substance use disorder (SUD) in Nigeria.

### Methods

This was a community-based cross-sectional study design using qualitative data collection methods. Data was obtained from the participants using a pre-tested focus group discussion (FGD) guide. Data was collected from three states in Nigeria including Anambra, Enugu and Nasarawa states. Twelve FGDs were conducted among 116 participants who were beneficiaries of the integrated service delivery for TB/MH/SUD and their relatives. There were four FGDs in each state. The discussions were conducted separately for patients and relatives and for male and female participants. QDA Miner Lit v2.0.6 was used in the thematic analysis of data.

**Data availability statement:** All relevant data are within the paper and its Supporting Information files.

**Funding:** STOP TB Partnership. (Reference, STBP/TBREACH/GSA/W10-10213).

**Competing interests:** The Authors have declared that no competing interests exist.

**Abbreviations:** FGD, Focus group discussion; HIV, Human immunodeficiency virus; ISD, Integrated service delivery; LGA, Local government area; LMIC, Low-and middle-income countries; MH, Mental health; MDR TB, Multi-drug resistant tuberculosis; PHC, Primary health care; QDA, Qualitative data analysis; RR TB, Rifampicin-resistant tuberculosis; SUD, Substance use disorder; TB, Tuberculosis; WACP, West African College of Physicians; WHO, World Health Organization.

## Results

Almost all the participants perceived the three disease entities as being linked to one another hence it may not be the best to manage each condition in isolation thus supporting the integrated approach. The participants noted the positive provider attitude of the healthcare workers involved in the program. They were of the opinion that integrated service delivery (ISD) has improved the awareness of the three disease entities among the populace. Most of the participants expressed their willingness to patronize integrated service delivery at the community level from trained lay health workers. This willingness to patronize was predicated on the approval of the program by the government.

## Conclusions

Positive provider attitude of health service providers will be a good boost to efforts to improve health service delivery in Nigeria including integrated service delivery. Government has a key role to play in community acceptance of health service delivery programs. The program increased the awareness of the three diseases among the people. Thus, increasing the community awareness of TB, mental health and substance use disorders should be prioritized. Adopting the integrated service delivery approach will be of value. Consideration should be made on the use of lay health workers for the delivery of such services at the community level especially in rural areas. There is a need to incorporate community perspectives on the value, benefits, barriers and acceptability of integrated service delivery into policies guiding TB/MH/SUD integration in Nigeria.

## Background

The World Health Organization (WHO) defines integration of health services as 'the management and delivery of health services so that clients receive a continuum of preventive and curative services according to their needs over time and across different levels of the health system' [1]. The concept of integrated service delivery was the focus for the envisaged "essential health care under one roof". It was also the main reason for the attention paid to primary health care (PHC) in the 1980s [1]. It is regarded as the approach of combining services of multiple interrelated diseases in such a way as to increase overall efficiency of the health system and at the convenience of the patients. This may explain why 'Pillar one of the WHO End TB Strategy' focuses on integrated patient-centered care and prevention including action on TB and comorbidities [2].

Tuberculosis (TB) is the leading cause of death from a single infectious agent [3]. For example, in the year 2023 an estimated 10.8 million people fell ill with TB globally. The total number of people newly diagnosed with TB across the globe increased from 7.5 million in 2023 to 8.2 million in 2024 [3]. Nigeria has the highest TB burden in Africa and accounts for 4.6% of estimated individuals that developed TB globally.

Nigeria and 29 other countries are regarded as high TB burden countries. This is because these countries contributed about 87% of all people that developed TB globally [3]. Nigeria is also included in the list of 14 countries that are in all the three WHO global high burden country lists for TB, TB/HIV and MDR-TB [3]. This is because over 34,000 people with TB are also infected with HIV and 2061 individuals are MDR/RR-TB confirmed cases [4].

Mental health conditions among TB patients have been found to range from 22% to 84% in various countries [5]. Specifically, the prevalence of depression among TB patients is in the range of 9% to 84% [5]. Mental health has been described as a state of well-being in which every individual realizes their own potential, copes with the normal stresses of life, works productively and fruitfully and is able to make a contribution to their community [6]. An estimated 85% of individuals with mental illness in low-and middle-income countries (LMIC) receive no form of treatment for mental health [7]. In Nigeria, an approximate 20% of inhabitants have mental illness, the most common being depression and anxiety disorders [8]. The awareness of mental disorder is very low in Nigeria [9]. The commonest cause of mental health disorders in Nigeria is attributed to substance abuse [9].

Substance abuse is the harmful use of psychoactive substances including alcohol and illicit drugs [10]. The WHO estimates that in the year 2021, a total of 296 million people aged 15–64 years had used psychoactive substances, and 39.5 million people could be affected by drug use disorders [11]. The United Nations Office on Drug and Crime in Nigeria revealed that 14.4% of individuals aged between15 and 64 years abuse drugs [12]. A descriptive national survey of drug use in Nigeria revealed that the pattern of drug use is different across the six geo-political zones of the country. Findings from that study showed that alcohol had the highest lifetime prevalent use of 30.3% among individuals aged 15–64 years old while cannabis was the most commonly used illicit drug with a lifetime use of 6.6% [13]. There is evidence that individuals diagnosed with mental illnesses are at an increased risk of developing substance use disorders and vice versa [14].

Integrated service delivery has been adopted into the Nigeria health system, and this has helped to increase access to service and coverage especially in maternal and child health programs including immunization services. Another example of integration is the combined TB and HIV service delivery which contributed to the reduction of TB burden among persons living with HIV and minimizing the risk of HIV infection in presumptive and TB diagnosed patients [15]. Other efforts towards integration include the TB/diabetes mellitus comorbidity management [16,17]. Furthermore, the integration of services helps to promote continuity of care, reduce stigma and also, improve access to services especially for vulnerable populations [18].

There has been a proposition to integrate mental health into primary health care [19]. Another such opinion is the need to integrate mental health education into all healthcare programs. This was viewed as a way to increase awareness of community members on mental health [20]. It has also been observed that identifying the relationship between mental health and substance use disorders will guarantee the development of effective interventions and policies that will ultimately promote holistic well-being [21]. This is because delivering the services in isolation will result in fragmented care delivery and missed opportunities for early intervention. Integrated service delivery particularly suits the local Nigerian context since funding for single disease programs is unsustainable necessitating the need to adopt cost-effective approaches bearing in mind the high burden of TB, mental illness and substance use disorder in the country. Furthermore, integration of health service delivery is associated with lower cost and improved treatment outcomes [22]. It also enhances equity [23]. This study was designed to examine community perspectives on integrated service delivery for TB, MH and SUD in Nigeria.

## Methods

### Study setting

Nigeria is made up of 36 states and the Federal Capital Territory. The states which serve as the second tier of government are distributed among the six geo-political zones of the country. This research took place in three states, two of which are in southeast geo-political zone, (Enugu and Anambra states) and one state in north-central geo-political zone, (Nasarawa

state). Nigeria has a total of 774 local government areas (LGAs) which are the third tier of government. The study was conducted in 10 local government areas; four are in Anambra state (Awka North, Awka South, Nnewi North and Onitsha South); another four in Enugu state (Awgu, Enugu East, Enugu North and Igboeze North) while two LGAs (Eggon and Keffi) were included from Nasarawa state. The three states were selected because they have similar TB case notification rates and the availability of tertiary health institutions for the referral and management of issues related to mental health. Anambra and Enugu states are in the tropical rain forest while Nasarawa state lies in the Guinea savannah. Igbo language is the main local language in Enugu and Anambra states while in Nasarawa state, the common local languages include Agatu, Basa, Eggon and Gbayi.

In the current standard of care, the patient pathways for TB and MH/SUD services in Nigeria are unlinked and totally independent. While the siloed delivery of TB services is provided at community and primary, secondary and tertiary facility levels in all the states, only specialized entry points like psychiatric hospitals and psychological medicine departments in tertiary health facilities currently offer MH/SUD services where all types of mental health disorders are taken care of. In the context of this study, integrated service delivery was provided at the health facility and community levels. In the health facility level, it meant screening all presumptive TB cases for MH/SUD and also screening individuals presenting at MH/SUD clinics for TB. All the screenings were also carried out at the community level and all individuals who were positive for any of the three disease entities received treatment services.

### Study design, participants and sampling

This was a community-based cross-sectional study design using a qualitative data collection method.

### Study population

One hundred and sixteen individuals participated in twelve focus group discussions (FGDs). Half of the participants were patients who were beneficiaries of the integrated delivery of TB/MH/SUD services in the project states while the other half were their relatives and caregivers during the period of management. Four FGDs were conducted in each of the three states selected for the study. The FGDs were conducted separately for patients and their relatives and for each of the groups, there were separate FGDs for the male and female participants.

### Sampling

Purposive sampling was used for the selection of the participants for the study. Each of the selected beneficiaries of the integrated service delivery program was requested to come for the interview with the caregiver that accompanied him/her during the period he/she benefitted from the program. None of the participants selected for the study declined to participate.

### Study instrument and data collection method

Information from the participants was obtained with the use of a pre-tested FGD guide which included questions on the advantages and disadvantages of integrated service delivery and their acceptability of community-based integration by lay health workers. The pre-testing was done in a community not selected for the study. The intention in pre-testing the study tool was to identify and correct ambiguities in the study instrument where they do exist. The FGDs were conducted using English language or the prevailing local language in the community as the case may be. The discussions took place in public places in the communities including town halls and primary schools. All the discussions were recorded digitally. A note taker was also available to take notes manually. Healthcare workers working in the facilities that were used for the integrated service delivery and community guides assisted the research team in the selection of the participants for the study. They related between the research team and the participants all through the study and were also involved in the

selection of a suitable date for the discussion. Follow-up questions using probes were applied during the FGDs as way of obtaining more detailed explanations from the participants. The average duration of the discussions was 55 minutes. No FGD was repeated. Data saturation was reached during the FGDs as no new information was obtained.

Two Authors moderated the FGDs. Both are Fellows of the West African College of Physicians (WACP), Faculty of Community Health. They are both Senior Lecturers, one at Ebonyi State University Abakaliki, Nigeria while the second is a staff of Alex Ekwueme Federal University Ndufu-Alike, Ikwo, Ebonyi State, Nigeria. Both are male and have some experience in qualitative studies [24–27].

## Data management

The recorded discussions of FGDs were transcribed verbatim after each session. In cases where the discussion was done using the local language, translation into English language was done. For quality assurance purposes, the scripts were compared with the written notes for completeness and accuracy. Then each script was checked against the audiotape by an independent reviewer. As a way of verifying the quality of transcriptions, the recordings were doubly transcribed. The scripts were then checked for similarity and where differences existed, these were reconciled by the transcribers. Data was analyzed thematically. Coding of transcripts was done by two of the researchers and based on pre-determined and emergent themes during the coding process. The pre-determined themes were derived from the study objectives. The process of thematic analysis included familiarization with data, generating initial codes, searching for themes, reviewing, defining and naming themes and report writing [28]. QDA Miner Lite v2.0.6 was used in the analysis of the data.

## Ethical consideration

Ethical approval for the study was obtained from the Health Research Ethics Committee of University of Nigeria Teaching Hospital Ituku Ozalla, Enugu, Nigeria (Reference number, NHREC/05/01/2008B-FWA00002458–1RB00002323). The recruitment of participants and data collection for the study took place from August to October 2024. Before the participants were recruited into the study, they were required to sign a written informed consent form. The nature of the study, its relevance and the level of their involvement were made known to them. Participation in the study was voluntary, and participants were informed that they were free to withdraw from the study even after giving consent to participate. The participants were also informed and assured that information provided during the discussion is strictly for research purposes and will be kept confidential and treated anonymously.

# Results

## Participants' profile

The age of the participants ranged from 29 to 61 years with a mean age of 45 years. Half of the participants were male. Half of the participants were patients treated during the integrated delivery of TB/MH/SUD services while the remaining half were their relatives/caregivers. Most of the participants, 58% had secondary education. Most of the participants, 64% were self-employed.

## Themes

Three themes emerged from the FGDs including perceived value of integrated TB/MH/SUD service delivery, perceived benefits and barriers to integrated service delivery and the acceptability of community-based Integration through lay health workers.

### a.Perceived value of integrated TB/MH.SUD service delivery

Almost all the participants perceived integrated service delivery of TB/MH/ SUD to be a good program. This was based on the wonderful results they have seen from the implementation of the program and their understanding that the three

disease conditions are linked to one another hence it may not be the best to manage each condition in isolation. The participants largely opined that it made sense for the three health services to be delivered together in an integrated approach.

*'The three diseases do not occur in isolation, by taking care of the three diseases together you have given the person complete treatment'.* (Male/PT/03/06).

*'Most of the people that have TB, they do have mental health issues because of the marginalization that is associated with the disease. Some people if they find out you have TB, the way they even treat you will affect your mental health'.* (Female/PL/02/04).

*'What I like the most in this combination (referring to ISD) is that TB may lead to mental disorder, mental disorder has a connection with addiction. So, if you place the diseases properly, it helps to explain that you cannot separate TB from mental health and substance abuse disorder'.* (Male/PT/02/07).

There were some remarkable comments about the health workers and their activities during the period of the integrated service delivery. The participants affirmed the good provider attitude of the healthcare workers that were involved in the implementation of TB/MH/SUD service delivery. These were how the participants exemplified their thoughts:

*'I like the program and it is acceptable to me because of the counselling. I was a drug addict but have stopped. I will continue to go to the health facility so that I can know more about the effects of drug abuse & mental disorder. The counselling I received from the health workers changed my life for good.'* (Male/PT/01/02).

*What made the service delivery good was the encouragement they (referring to the healthcare workers) provided. Like me, when I started my own treatment, I thought there was no hope, but they kept on encouraging me. Through their help, I got better and today, I am well.'* (Female/PT/03/09).

One of the participants who was pleased with the integrated service delivery pleaded that the services, especially the management of mental health disorder should be extended to all primary health centers in the state.

**b.Perceived benefits and barriers to integrated service delivery**

**bi. Benefits**

Most of the participants were of the impression that the integrated service delivery had more advantages than disadvantages. The major advantage of the integrated service delivery as expressed by the participants was the reduction in cost especially that of transportation. Also, a reduction in stress of moving from one health facility to another especially when one is seriously ill. There was a remark that stress is also part of mental health disorder. It was also said to save time. These were how the participants made their views known.

*'The benefit is first of all less expenses on transportation and less stress because moving from one hospital to another will add more stress to the person who is sick already. Then, staying in the same hospital to treat three diseases is like using one stone to kill three birds.'* (Female/PT/03/03).

*'We prefer to receive the treatment for the three diseases in one place because a seriously ill patient will not be able to move from place to place accessing isolated services for TB, mental health and substance use disorder.'* (Male/PT/01/10).

*'The advantage is that it saved time and resources receiving the treatment in one place instead of going to three different places. If you receive treatment in one place, it saves time and also resources because of transportation.'* (Male/PL/02/21).

The next advantage from the views of the participants was that all the treatment they received in the integrated service delivery approach was free.

*'It is free treatment for the treatment of three diseases in one place. It is a good thing and I am happy about it. The only cost that we had to bear was that of transportation only.'* (Male/PT/03/14).

There were observations among the participants that the integrated service delivery has increased the awareness of tuberculosis, mental health and substance use disorders among the people. This was seen as a way of reducing the spread of the diseases.

*'Another advantage is the increase in public awareness of these three diseases. This approach has made us know that these three disease conditions could exist together. So, the increase in public awareness is one of the advantages and when you have good public awareness of a disease, the rate of occurrence of the disease condition will be reduced'.* (Male/PT/03/12).

There was the affirmation of good treatment outcome from the integrated service delivery approach. One of the participants viewed the integrated service delivery as a form of balanced diet.

*'One is that it will gave good results because the diseases were followed up one after the other. It's like what is called balanced diet. Because when you are able to get treated for this, this and that, at the same time you get maximum benefit and result.'* (Male/PL/02/19).

The participants also viewed the integrated service approach as an opportunity for being screened for other disease conditions which the person may not be aware of.

*'The arrangement gave one the opportunity of being tested for the three disease conditions at the same time. For example, some people had TB and mental health issues at the same time without knowing it.'* (Female/PL/01/16).

Based on the observation that the three diseases are linked there was agreement among the participants that the treatment of the three disease conditions should not be in isolation.

*'If one goes to different places for the treatment, the health workers will manage mental health in isolation and tuberculosis in isolation so the linkage will not be there. But when the individual is managed in the same place, the health workers will be able to link the three diseases together and make a better treatment'.* (Male/PT/03/11).

Two of the participants noted that even if the patients are managed by different doctors in the same health facility, that the doctors will put their knowledge together for the good of the patient. Thus, it was concluded that the patients will recover faster if integrated service delivery is in place.

**bii. Barriers**

Even though a number of participants insisted that the integrated service delivery had no disadvantages, some of the participants expressed some disadvantages. Most of the participants who spoke on the disadvantages were concerned about the safety of persons with mental health disorder since TB is infectious. This was how one of the participants made his views known:

*'Mental health is not an infectious disease but tuberculosis is infectious. If the individuals that have the two diseases are treated in one place, someone with mental health might be infected with tuberculosis which is now a disadvantage.'* (Male/PL/03/09).

One of the participants also pointed out that someone with TB moving from one health facility to another perhaps for treatment of mental disorder could also spread TB infection to others. Some of the participants were concerned about the specialization of the medical personnel who will be attending to people with TB, mental health and substance use disorders. There was a remark that each doctor should focus on his own specialty instead of being jack of all trade. It was said that in a teaching hospital, each doctor stays in his/her department and takes care of people in his/her own chosen field.

*'There are doctors that treat TB and may not treat one with mental health condition and doctor that treats a mental disorder cannot see someone with eye problems and treat it, so everyone has an area of specialization. So, let everyone stay in his own side, let them separate i. If you are depressed you go to doctor for depression and if you are suffering from TB, the doctor for TB will take care of you.'* (Female/PT/02/06).

A few of the participants who spoke on the disadvantages of integrated service delivery pointed out that it increased the waiting time in the health facilities.

*'The disadvantage is that this approach increased the time spent in the health facilities unlike before and some people do not want to stay long when they come to the hospital, they will say they are wasting their time in the queue. Iif the services are provided in different places, if you have this condition you can go to this place and if you have the other condition you can go to the other place that will ensure that people do not waste time in the health facilities.'* (Female/PL/01/18).

In the light that there may be more work for the healthcare workers based on the integrated service delivery approach which may lead to an increase in waiting time, one of the participants who supported the integrated service delivery recommended that the government should employ more healthcare workers so as to reduce the waiting time in the hospitals designated for such services.

It is important to point out that the perception of mental health by the participants influenced their decision on integrated service delivery. The participants who viewed patients with mental health disorder as 'mad' or 'one whose head is not good' wondered how such group of people should be kept in the same area with TB patients. One of the participants pointed out that patients with mental health disorders may require to be chained and as such should be managed separately. However, the participants who viewed mental health differently were of the opinion that integrated service delivery is ideal since most TB patients because of the disease may have mental health issues. There was also a position by the participants that a good mental health is necessary to achieve a good adherence to treatment.

### c.Acceptability of community-based integration through lay health workers

Most of the participants were in support of patronizing integrated service delivery at the community level from trained lay health workers. Most of them who expressed willingness to patronize the services of the lay health workers said they will do so only if the government is involved in establishing such treatment centers in the communities. The participants were sure that if the government is involved in the program, the lay workers will be adequately trained and equipped with the requisite laboratory tests and medicines. They were certain that government would create the needed awareness among the people of the existence of such treatment centers. Furthermore, the services will be free of charge or at least affordable. There was an affirmation that the government cannot harm its own citizens and that providing such services at the community level is the responsibility of government and also the right of the citizens. These were how the participants expressed their views:

*'Such a community-based program is good because it is a thing like this that government should do for the people in rural communities because if you go there, you will see that the people are really suffering there. Some of them don't even have good drinking water. So, it is good because it will help them. It is a good idea.'*(Female/PL/01/07)

*'Government will not do harm to anyone and it is also our right as citizens to have access to healthcare. Anything that government is doing will be of good quality, the drugs that they will provide will be for the benefit of the members of the community. So I will go there for treatment.'* (Female/PT/02/09).

The next major reason for patronizing the community lay health workers by the participants is that it is an opportunity to bring such services closer to the people and that will help save lives, time, transportation cost and stress among the rural dwellers.

*'What makes it good is that, it will save time, if they decide to treat the diseases in the rural communities. It will also save the people the stress of moving to the urban areas every time they are sick.'* (Male/PT/ 03/15).

*'Yes, the idea is good, because anyone that knows that they have such a facility in the village, will go to the village. You stay there and take your medication; it will save you the stress of coming down to the township, it will save you the cost of transportation and stress.'*(Female/PL/01/14).

One of the participants made known the steps he will take to find out if the lay health workers are trained before he will patronize their services. He said this bearing in mind his experience as someone who has been treated for TB.

*'I will go the community center. My reason is that, when I get there, if I tell them what is wrong with me and if they bring out drugs to give me, I will tell them no— they should do a lab test. When they do the lab test, I will know that they know what they are doing. If they say no and insist that I should take the drug, if I collect the drug, I will throw it away on the road'.* (Male/PT/03/07).

Two of the participants were of the opinion that the government should provide the lay workers with uniforms or aprons like the ones worn by individuals who take part in house-to-house immunization services. The impression is that the uniform or apron will make the community members know that they are sent by the government and they have received the necessary training to enable them to perform their duties.

*'They should be on uniform even if it is an apron like these people that go to give immunization. Then we can say it is government that sent them. If they did not give them uniform or apron maybe people will not allow them to treat their children.'* (Male/PL/03/10).

One of the participants suggested that the duty of the lay health workers should be to help create awareness of the three diseases including TB, mental health and substance use disorders among the people and not to offer treatment services to individuals.

The few participants who have in mind not to patronize the services of the lay health workers said they will not go there because they are not doctors.

*'It is not a good idea because they are not doctors, they didn't go to any school. If they give you medicine, it won't work; they may in a bid to give you drugs, give you an overdose.'* (Male/PT/03/08).

Another concern raised by the participants is why the government will not use the opportunity of the community-based program to employ health workers instead of using lay health workers.

*'There are many health workers looking for work, why not pick them instead of using lay health workers. The health workers went to school but are neglected and their job given to others to do. Why? Let health workers do their job. For me, if anyone didn't go to school, I will not allow him to treat me.'* (Male/PT/01/09).

Two of the participants pointed out that the lay health workers will collect money from the people even if the government instructs them not to do so. One of the participants said when a community person is involved in matters related to health in the same community, they tend to focus on their own family members instead of everyone.

## Discussion

This study was designed to examine community perspectives on integrated service delivery for TB/MH/SUD in Nigeria. Three themes were identified which included the perceived value of integrated service delivery which highlighted the positive attitude of the service providers. Other themes included perceived benefits and barriers to integrated service delivery and the acceptability of community-based integrated service delivery by lay health workers of which most of the participants expressed their willingness to patronize such services.

Almost all the participants were of the opinion that the three disease conditions including TB, mental health and substance use disorders are linked to one another. This finding is of relevance coming from the beneficiaries of the program and their relatives. It portrays a good understanding of the integrated service delivery process. There is evidence of the good impact of integrating mental health into TB services. For example, a study in Zambia advocated for the integration of mental health training in TB services as a way to change the negative attitudes about mental health among TB service providers [29]. This suggestion became necessary based on the observation that TB stakeholders and providers in that country had a poor understanding of mental health and illness which did not support the mental health needs of TB patients. As a practical demonstration of the good in integrated service delivery, a feasibility assessment of integrating depression care into TB services in Pakistan revealed a high burden of depression among TB patients [30]. Similarly, when mental health service was integrated within TB programs in Pakistan, the approach helped to reduce symptoms of depression and anxiety while also improving TB treatment completion [31].

The participants in this study were full of commendations for the healthcare workers that took part in the delivery of services during the implementation of the integrated service delivery program. This is laudable bearing in mind the effects of negative provider attitude to the TB program. For instance, the results of a study among pulmonary TB patients in Osogbo, Nigeria revealed that perceived negative provider attitude was associated with TB diagnostic delay [32]. Similarly, a qualitative meta-analysis of facilitators and barriers to TB diagnosis and treatment in Nigeria identified attitude of TB service providers as one of the barriers to diagnosis of TB in the country [33]. A similar result was also obtained from the result of a study in another African country. For example, a study in Zambia identified negative provider attitude as one of the health system software elements that slow referral mechanisms and also delay TB notification [34]. Elsewhere in Ukraine, it was found that TB providers being unsupportive was a major challenge to adherence to treatment [35]. It could be said that the training the health workers received before and during the implementation of the integrated service delivery program accounted for their good provider attributes and this should be sustained

The major advantage of the integrated service delivery as expressed by the participants was reduction in cost especially that of transportation. This speaks volume of how the people perceived the immense benefits of the integrated service delivery program. Having understood that mental health and substance use disorders are closely linked to TB, they imagined that what could have necessitated referral and subsequent visits to perhaps psychiatric hospitals were handled in the same health facility. Another concern was the possibility of even diagnosing the mental health disorder if the integrated service delivery was not available. Thus, they viewed the integrated service delivery as saving them transport costs and other logistics since almost all psychiatric hospitals in Nigeria are located in urban areas. Another observation by the participants was that all the treatment they received during the course of the program was free. In Nigeria, all matters

related to diagnosis and management of TB are free and same were applied to all the patients that presented in the designated health facilities all through the duration of the integrated service delivery for TB, MH and SUD.

An important observation by the participants is that the integrated service delivery program increased the awareness of the people of the three disease entities including TB, MH and SUDs. This finding is of importance bearing in mind how the people perceive the three diseases included in the program. For example, a study among urban slum dwellers in Lagos, Nigeria revealed a poor knowledge of symptoms and curability of TB and also the fact that TB treatment in Nigeria is free [36]. Bear in mind that the participants in this study recognized the free treatment they received for TB/MH/SUD during the course of the program. When it relates to mental health, a study among in-school adolescents in southeast Nigeria revealed a poor knowledge of mental illness [37]. Another study in South Africa identified increased mental health awareness as one of the measures to enhance the implementation of integrated mental health, TB and maternal health services [38]. Also, a study among adolescents and young adults in southwest Nigeria revealed that none of the respondents were aware of existing laws in the country on substance abuse [39]. Based on these observations and the findings from this study, increasing awareness of TB, MH and SUD among the populace should be of priority.

Even though, a few participants insisted that the integrated service delivery has no disadvantages, most of the participants who spoke on the disadvantages were concerned about the safety of persons with mental health disorder since TB is an infectious disease. This again stems from the understanding of the participants that TB is a communicable disease. Some of the participants were also concerned about the specialization of the medical personnel or perhaps the competence of healthcare workers to attend to people with TB, MH and SUDs at the same time. This portrays how conscious the average Nigerian is when it concerns medical specialization. For example, a study among enrollees in northeast Nigeria revealed that majority of the respondents chose a tertiary hospital as their care provider because of the availability of specialist services [40]. Also, one of the reasons for non-utilization of primary health services in a rural community in southeast Nigeria is the unavailability of medical doctors in primary health centers [41]. This could explain the concern of the participants as they were wondering if there will be enough healthcare workers with the expertise needed to combine the treatment for TB, MH and SUD.

A few of the participants who spoke on the disadvantages of integrated service delivery were of the opinion that it will lead to an increase in waiting time in the health facilities. This is expected based on the exigencies of integrated service delivery. In a study in Benue state, Nigeria on integrating neglected tropical diseases with mental health services, one of the major concerns of the participants was an increase in waiting time [42]. The result of a systematic review on integrating TB and non-communicable diseases in low-and middle income countries identified increased workload on the part of providers as one of the barriers to the program [43]. This increased provider workload could be what the patients interpreted as increased waiting time from the results of this study.

From the results of this study, most of the participants were in support of patronizing integrated service delivery at the community level provided by trained lay health workers. This was irrespective of the fact that some of the participants were concerned on the specialization of the medical personnel or health workers who will be attending to people with TB, MH and SUDs at the same time. Perhaps, the acceptance of the suggestion may be because it was centered on rural communities where access to healthcare services is not as good as that in the urban areas of the country. This view of the participants is in tandem with previous scientific submissions. For example, a Cochrane systematic review concluded that lay health workers provide promising benefits in improving TB treatment outcomes when compared with usual care [44]. Another study in Pakistan affirmed that with adequate training, lay counsellors could efficiently handle diverse mental health problems in primary health care settings [45]. It is also important to point out that the willingness of the participants to receive integrated service delivery at the community level provided by trained lay health workers centered on the approval of the government for such services. Approval by government was reported as one of the facilitators to integrating depression care in TB services in South Asia [46].

A study in India concluded that trained lay members of the community could be a vital resource in delivering mental health care in resource-poor settings [47]. There was a position that the utilization of the services of lay health workers is possible but will require training and supervision [47]. This was in line with the position of the technical advisory group of the WHO which made a case for person-centered and community-based mental health services in response to the far-reaching mental health impact of COVID-19 pandemic [48]. Prior to that there has been a postulation that with the shortage of mental health experts globally and the increasing burden of mental illness that utilizing peer workers and other non-professionals may be the only solution to providing mental health services in both low- and high-resource settings at least in the short term [49]. Moreover, a comprehensive review of mental health services across selected countries in sub-Saharan Africa revealed that community-based models of care coupled with advocacy are important for reducing stigma and promoting sustainable mental healthcare in the region [50]. The results of a study in Pakistan went further to attest that such community-based models will not only reduce the stigma associated with seeking mental health care outside the home environment but will reduce the high cost associated with such practices [51]. This necessitates the need to initiate community-based mental health services in rural communities and the phased and gradual adoption of integrated service delivery of TB, MH and SUD in Nigeria.

The strength of this study lies in the fact that this was the first time an integrated service delivery program involving TB/MH/SUD was initiated in Nigeria. The focused attention on the program by the initiators of the project including wide stakeholder engagements, training and re-training of healthcare workers and supportive supervision may have helped to bring about the many benefits of the program. For example, the providers' positive attitude improved patients' experiences of integrated service delivery. Suffice it to say that if such program is now deployed at the national level without such close attention as enumerated above, the benefits may not be the same as obtained in this study. In any case, bearing in mind the numerous gains of the program, a phased implementation at the national level will be of good. Another limitation of the study is the possibility of social desirability bias on the part of the participants who may perceive the data collection process as an audit exercise. Efforts were however made in explaining to the participants that data collected during the FGDs were for research purposes only.

## Conclusion

Positive provider attitude of health service providers will be a good boost to efforts to improve health service delivery in Nigeria including integrated service delivery. Government has a key role to play in community acceptance of health service delivery programs. The program increased the awareness of the three diseases among the people. Thus, increasing the community awareness of TB, mental health and substance use disorders should be prioritized. Adopting the integrated service delivery approach will be of value. Consideration should be made on the use of lay health workers for the delivery of such services at the community level especially in rural areas. There is a need to incorporate community perspectives on the value, benefits, barriers and acceptability of integrated service delivery into policies guiding TB/MH/SUD integration in Nigeria.

## Acknowledgments

The Authors remain grateful to d the beneficiaries of the TB/MH/SUD program for their participation in the study.

## Author contributions

**Conceptualization:** Martin Njoku, Charles Nwafor, Chinwe Eze, Okechukwu Ezeakile, Anthony Meka, Ngozi Ekeke, Iyama Francis, Daniel Egbule, Joseph Chukwu, Charles Esekhaigbe, Chijioke Osakwe, Daniel C. Oshi, Ngozi Murphy-Okpala.

**Data curation:** Martin Njoku, Charles Nwafor, Chinwe Eze, Okechukwu Ezeakile, Anthony Meka, Ngozi Ekeke, Daniel Egbule, Joseph Chukwu, Edmund Ndudi Ossai, Chibuike Agu, Grace Bernard-Asadu, Clement Adesigbin, Obioma Chijioke-Akaniro, Daniel C. Oshi, Ngozi Murphy-Okpala.

**Formal analysis:** Chinwe Eze, Ngozi Ekeke, Iyama Francis, Daniel Egbule, Charles Esekhaigbe, Edmund Ndudi Ossai, Chibuike Agu, Ngozi Murphy-Okpala.

**Funding acquisition:** Martin Njoku, Charles Nwafor, Chinwe Eze, Anthony Meka, Daniel Egbule, Ngozi Murphy-Okpala.

**Investigation:** Martin Njoku, Charles Nwafor, Anthony Meka, Ngozi Ekeke, Iyama Francis, Daniel Egbule, Joseph Chukwu, Charles Esekhaigbe, Chijioke Osakwe, Chibuike Agu, Chukwuma Anyaike, Clement Adesigbin, Obioma Chijioke-Akaniro, Tunde Ojo, Daniel C. Oshi, Beatrice Kirubi, Jacob Creswell, Ngozi Murphy-Okpala.

**Methodology:** Martin Njoku, Charles Nwafor, Chinwe Eze, Okechukwu Ezeakile, Anthony Meka, Ngozi Ekeke, Iyama Francis, Daniel Egbule, Joseph Chukwu, Charles Esekhaigbe, Chijioke Osakwe, Edmund Ndudi Ossai, Chibuike Agu, Grace Bernard-Asadu, Chukwuma Anyaike, Clement Adesigbin, Obioma Chijioke-Akaniro, Tunde Ojo, Daniel C. Oshi, Beatrice Kirubi, Jacob Creswell, Ngozi Murphy-Okpala.

**Project administration:** Martin Njoku, Chinwe Eze, Okechukwu Ezeakile, Anthony Meka, Ngozi Ekeke, Iyama Francis, Daniel Egbule, Joseph Chukwu, Charles Esekhaigbe, Chijioke Osakwe, Beatrice Kirubi, Ngozi Murphy-Okpala.

**Resources:** Martin Njoku, Charles Nwafor, Anthony Meka, Daniel Egbule, Joseph Chukwu, Obioma Chijioke-Akaniro, Daniel C. Oshi, Beatrice Kirubi, Jacob Creswell, Ngozi Murphy-Okpala.

**Software:** Charles Nwafor, Joseph Chukwu, Jacob Creswell.

**Supervision:** Martin Njoku, Charles Nwafor, Chinwe Eze, Okechukwu Ezeakile, Anthony Meka, Ngozi Ekeke, Iyama Francis, Daniel Egbule, Charles Esekhaigbe, Chijioke Osakwe, Edmund Ndudi Ossai, Grace Bernard-Asadu, Chukwuma Anyaike, Clement Adesigbin, Tunde Ojo, Daniel C. Oshi, Ngozi Murphy-Okpala.

**Validation:** Martin Njoku, Okechukwu Ezeakile, Joseph Chukwu, Charles Esekhaigbe, Ngozi Murphy-Okpala.

**Writing – original draft:** Edmund Ndudi Ossai.

**Writing – review & editing:** Martin Njoku, Charles Nwafor, Chinwe Eze, Okechukwu Ezeakile, Anthony Meka, Ngozi Ekeke, Iyama Francis, Daniel Egbule, Joseph Chukwu, Charles Esekhaigbe, Chijioke Osakwe, Edmund Ndudi Ossai, Chibuike Agu, Grace Bernard-Asadu, Chukwuma Anyaike, Clement Adesigbin, Obioma Chijioke-Akaniro, Tunde Ojo, Daniel C. Oshi, Beatrice Kirubi, Jacob Creswell, Ngozi Murphy-Okpala.

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
