## [Decision Letter · Decision Letter 0]

6 Oct 2025

Dear Dr. Ossai,

Thank you for submitting your manuscript to PLOS ONE. After careful consideration, we feel that it has merit but does not fully meet PLOS ONE’s publication criteria as it currently stands. Therefore, we invite you to submit a revised version of the manuscript that addresses the points raised during the review process.

We look forward to receiving your revised manuscript.

Kind regards,

Daniel Chukwuemeka Ogbuabor, Ph.D., M.D.

Academic Editor

PLOS ONE

Journal Requirements:

“STOP TB Partnership. (Reference, STBP/TBREACH/GSA/W10-10213).”

3. We note that your Data Availability Statement is currently as follows: All relevant data are within the manuscript and in Supporting Information files.

Additional Editor Comments:

1. Title:

Since acceptability is one out of three main themes explored in this study, limiting the title to acceptability would seem restrictive. Equally, the title should include the study type. While “Integration in Action” is apt, the study did not evaluate the implementation of the integrated service delivery (ISD) of TB/MH/SUD. Evaluations of such intervention would assess the fidelity of implementation, which truly assesses integration in action. Consequently, “Community perspectives on integrated service delivery for tuberculosis, mental health and substance use disorder in Nigeria: a qualitative study” will be a more suitable title.

2. Abstract

a. The first two sentences of the abstract do not rationalize the study. What gap does this study fill? The authors should consider this revision: “Despite being the central focus of essential health under one roof and a crucial principle of primary health care, relatively little is known about community perceptions on integrated service delivery in low-and middle income countries.”

b. In first sentence of the method’s section, ‘qualitative’ is missing in the study design.

c. The author should state the method of qualitative data analysis adopted in the study. I presume it is thematic analysis, but it must written explicitly. QDA is merely a tool.

d. The result section does not capture the key findings across the three themes found in the study. See my detailed comments on the main results section. Still, the authors must revise the statement: “the participants applauded the positive provider attitude of the healthcare workers in the program.” The sentence reads like news report! The true meaning of the statement reflects how providers’ attitude improved patients’ experiences of integrated service delivery.

e. Conclusion: “Positive provider attitude of health service providers will be a good boost to efforts to control TB in Nigeria” and “Adopting the integrated service delivery approach will be of value and this could be done in phases” are not based on the findings of the study. The authors did not study TB control in Nigeria; the study focused on ISD. There is no study finding on adopting ISD in phases. In my opinion, the conclusion should highlight the policy implications of the findings. The main conclusion is that community views matter in the design and implementation of integrated service delivery for TB/MH/SUD. There is a need to incorporate community perspectives on the value, benefits, barriers and acceptability of ISD into policies guiding TB/MH/SUD integration in Nigeria.

3. Introduction: The introduction requires a major revision in content and English language editing. There are many examples of redundant phrases and clauses throughout the manuscript such as “There has been a suggestion”, This is viewed as a way to”, and “It has also been observed…” Furthermore, the authors should minimize the use of the pronoun “it”.

a. Lines 86-87: Provide the reference (citation).

b. Move Lines 115-118 to the first paragraph of the introduction to sharpen the introductory paragraph.

c. The introduction failed to rationalize the study using pertinent literature. Lines 109-113 should be replaced with a review of the pertient literature on TB/MH/SUD service integration. The existing literature should highlight what we already know about value, benefits, barriers, and acceptability of integrated TB/MH/SUD service delivery.

d. The last paragraph of the introduction should summarize the knowledge gaps from the existing literature including paucity of empirical literature on the study topic globally, geographical gaps (few studies in the study setting), methodological gaps (study designs, data analysis, etc), and population gap. The authors must state the specific gap the current stuy fills, the purpose of this study, and the significance of the study (who will use the evidence and how?).

4. Methods

a. Study setting: Why did you select the three states? Perhaps, they were the project states for the ISD for TB/MH/SUD.

b. The authors should describe the ISD for TB/MH/SUD project under the study setting.

c. Study design, participants, and sampling: The authors should separate this sub-section into three different sub-sections to avoid missing important details of any part.

i. Lines 153-154: insert qualitative into the sentence just before study design.

ii. Participants: The subsection should appropriately titled “study population.’ The authors should describe the population from which the participants were sampled.

iii. Lines 154-159 describe a data collection procedure and should be moved to the data collection sub-section (lines 165-177).

iv. Lines 161-162: Move the sentence to the data collection sub-section.

v. Lines 175 and 177: change interview to FGD.

d. Data management:

i. Explain how the study derived the pre-determined themes.

ii. How many people coded the transcripts?

iii. The study seemed to have done thematic analysis but fell short of stating so. The authors should describe the process of thematic analysis.

5. Results

a. Theme 1: has three sub-themes including ISD as a worthwhile approach, positive providers’ attitude, and context of PHC.

b. Theme 2: sub-themes include advantages and barriers

i. Advantages include saves cost of transportation, reduces stress, saves time, affordable cost of care, increased awareness of the three diseases, and improved patient outcome/experiences.

ii. Barriers include mentally ill patients exposed to TB infection, lack of specialized care, increased waiting time, and TB patients risk attack from mentally ill patients.

c. Theme 3: Sub-themes include acceptance of ISD by LHWs, government approval, affordability, proximity, training of LHWs, provision of uniforms, clear roles for LHWs, employment opportunities, and informal payments to LHWs.

d. The authors must be clear about the unit of analysis. Is the unit of analysis individuals or FGDs? The usefulness of FGDs lies in the convergence or divergence of views within and across groups and states.

e. The above summary of the themes and sub-themes can be presented in a Table.

f. The authors must be intentioanl about the choice of verbs when describing participants’ perspectives. Such verbs like applauded, pleaded, and pleased should be avoided. Use stated, mentioned, claimed, noted, suggested, and similar verbs that are emotionally neutral.

6. Discussion

a. The discussion should have an introductory paragraph showing the purpose of the study, and the key findings warranting further exploration across the three thematic areas.

i. Perceived value of ISD – worthwhile approach to service delivery and positive provider attitude

ii. Perceived benefits and barriers to integrated service delivery – reduced transportation cost, increased awareness, safety of mentally ill people, and lack of expertise/specialization.

iii. Acceptability of community-based ISD through lay health workers – willingness to patronize ISD by LHWs, and training/capacity-building of LHWs.

b. Each paragraph should focus on one issue. However, Lines 483-500 (the fifth paragraph of the discussion) dealth with two topics – risk of mentally ill people contracting TB, and lack of medical specialization/expertise in the 3 diseases. The authors should consider separating them.

c. Lines 532-540: This is not a strength of the study but the strength of the ISD project. The authors must distinguish between the study (research process and its outcome) and the ISD project. The authors should explore strengths related to knowledge gap filled.

d. Curiously, the study omitted the limitations of the study.

7. Conclusion

The conclusion should restate the objective of this study and succinctly summarize the policy and practice implications of the main findings of the study. Avoid ambitious or extraneous recommendations that are not based on the findings.

Reviewers' comments:

Reviewer's Responses to Questions

**Comments to the Author**

1. Is the manuscript technically sound, and do the data support the conclusions?

Reviewer #1: Partly

Reviewer #2: Partly

2. Has the statistical analysis been performed appropriately and rigorously?

Reviewer #1: N/A

Reviewer #2: No

3. Have the authors made all data underlying the findings in their manuscript fully available?

Reviewer #1: No

Reviewer #2: Yes

4. Is the manuscript presented in an intelligible fashion and written in standard English?

Reviewer #1: Yes

Reviewer #2: Yes

Reviewer #1: Recommendation

The manuscript needs to be resubmitted with revisions

Overall evaluation

- Congratulations on the study highlighting the importance of integrated care for tuberculosis, mental health, and substance abuse, which is essential for effective tuberculosis control, especially in high-burden countries.

- Below are some comments to positively contribute to the enhancement of the study:

- Title: The title is appropriate, concise, and coherent with the study content. I recommend adding at the end of the title a colon followed by the specification of the study type, which will facilitate immediate identification of the research nature by readers.

- Abstract: The abstract is adequate, containing a clear synthesis aligned with the research objective.

- Introduction: The introduction covers relevant national and international concepts and data. However, the study would benefit from including a description of how tuberculosis care is provided in Nigeria, as well as access to health services in the country. In many countries, such care is offered in an integrated manner, especially driven by Primary Health Care, and international literature presents numerous promising examples that reinforce this integrated approach. It would be enriching to understand how Nigeria acts in disease control, considering that tuberculosis care still seems fragmented. Furthermore, given the study’s objective interrelating tuberculosis, mental health, and substance use as a chain perpetuating the disease, it would be pertinent to clarify how psychiatric services function in Nigeria, for example, whether they cover all types of mental disorders, including less severe cases, since it is mentioned that only hospitals fulfill this role. Although some methodological aspects briefly address this issue, we suggest that the introduction also include information about available mental health services and which substances are involved in substance abuse.

- Methodology: It is recommended to include the following information to enrich this section:

- In the section regarding data collection instruments, highlight that the instrument used was a semi-structured interview; specify which questions were used in the focus group discussions, even if based on prior pilot studies.

- In data management, inform how many researchers participated in the transcription, and whether any software was used for this step or if it was performed manually.

- Detail the technique used for qualitative data analysis. Although the use of QDA Miner Lite v2.0.6 software was mentioned, it is important to specify the analytical technique employed, whether thematic, discourse analysis, or other, and to detail it for replicability.

Results:

- I suggest removing the first paragraph of the results section and including this information in the methodology, adding that interviews were conducted by two researchers experienced in the topic; it is not necessary to name them.

- Regarding subject characterization, as the methodology mentions that 50% of participants were affected individuals and 50% were family members, it would be important to clarify the later percentages to avoid doubts about which groups they refer to. I recommend including a table detailing the sociodemographic profile of participants, appropriately segmented by group.

- About the selection of statements, it is important to mention in the methodology the analytical technique to understand how responses were selected, given that 12 focus groups with 116 individuals were used.

Discussion:

- It is recommended to incorporate international examples of successful integrated models to broaden the translation and applicability of the results. Additionally, a brief discussion of primary health care relating to the study findings would be relevant. From this perspective, adding examples of integrated approaches from countries with better disease control could further enrich the discussion.

Conclusion:

- Currently, the conclusion is somewhat generic. I recommend strengthening it based on the main findings of the study, highlighting lessons learned. It would also be important to include a section describing the research limitations, bringing transparency and foundation to the interpretations.

Reviewer #2: This manuscript is a good body of knowledge that will benefit from some minor revisions. Please see my comments below:

1. Background; Strengthen the justification for integration by highlighting gaps in siloed service delivery.

2. Study Setting

• Some details (ecological zones and local languages) feel overly descriptive for the Methods section. Unless these factors directly influenced data collection or interpretation, consider streamlining or moving them to the Introduction/Background.

3. Sampling Methodology

• The sampling approach is not clearly outlined. While purposive sampling is mentioned, the selection criteria ( age, diagnosis, treatment stage) should be specified.

• Clarify participant distribution: 116 participants across 12 FGDs, but this should be stated explicitly, including how patients and relatives were divided.

4. Study Instrument and Data Collection

• The narrative in this section can be tightened. Currently, there is some redundancy, particularly in describing the FGD process.

• The description of settings such as “public places like town halls and schools” should acknowledge possible limitations related to privacy, confidentiality, and risk of social desirability bias.

5. Data Management and Analysis

• The explanation of thematic coding is repetitive. Statements such as “coding of transcripts…based on predetermined and emergent themes” and “themes were reviewed and grouped under wider themes” can be streamlined for clarity.

.

Reviewer #1: **Yes:** Rosiane Davina da SilvaRosiane Davina da SilvaRosiane Davina da SilvaRosiane Davina da Silva

Reviewer #2: No

---

## [Author Response · Author response to Decision Letter 1]

23 Oct 2025

Response to Reviewers

Editor’s comment

Author response

Thanks for the information

Editor’s comment

“STOP TB Partnership. (Reference, STBP/TBREACH/GSA/W10-10213).”

Author response

Already included

Editor’s comment

3. We note that your Data Availability Statement is currently as follows: All relevant data are within the manuscript and in Supporting Information files.

Author response

Already included

Editor’s comment

Author response

Thanks

Editor’s comment

1. Title:

Since acceptability is one out of three main themes explored in this study, limiting the title to acceptability would seem restrictive. Equally, the title should include the study type. While “Integration in Action” is apt, the study did not evaluate the implementation of the integrated service delivery (ISD) of TB/MH/SUD. Evaluations of such intervention would assess the fidelity of implementation, which truly assesses integration in action. Consequently, “Community perspectives on integrated service delivery for tuberculosis, mental health and substance use disorder in Nigeria: a qualitative study” will be a more suitable title.

Author response

Study title reviewed in line with comments above. Many thanks.

Editor’s comment

2. Abstract

a. The first two sentences of the abstract do not rationalize the study. What gap does this study fill? The authors should consider this revision: “Despite being the central focus of essential health under one roof and a crucial principle of primary health care, relatively little is known about community perceptions on integrated service delivery in low-and middle income countries.”

Author response

Included

Editor’s comment

b. In first sentence of the method’s section, ‘qualitative’ is missing in the study design.

c. The author should state the method of qualitative data analysis adopted in the study. I presume it is thematic analysis, but it must written explicitly. QDA is merely a tool.

Author response

Revision made

Editor’s comment

d. The result section does not capture the key findings across the three themes found in the study. See my detailed comments on the main results section. Still, the authors must revise the statement: “the participants applauded the positive provider attitude of the healthcare workers in the program.” The sentence reads like news report! The true meaning of the statement reflects how providers’ attitude improved patients’ experiences of integrated service delivery.

Author response

Information noted

Editor’s comment

e. Conclusion: “Positive provider attitude of health service providers will be a good boost to efforts to control TB in Nigeria” and “Adopting the integrated service delivery approach will be of value and this could be done in phases” are not based on the findings of the study. The authors did not study TB control in Nigeria; the study focused on ISD. There is no study finding on adopting ISD in phases. In my opinion, the conclusion should highlight the policy implications of the findings. The main conclusion is that community views matter in the design and implementation of integrated service delivery for TB/MH/SUD. There is a need to incorporate community perspectives on the value, benefits, barriers and acceptability of ISD into policies guiding TB/MH/SUD integration in Nigeria.

Author response

Conclusion section has been revised

Editor’s comment

3. Introduction: The introduction requires a major revision in content and English language editing. There are many examples of redundant phrases and clauses throughout the manuscript such as “There has been a suggestion”, This is viewed as a way to”, and “It has also been observed…” Furthermore, the authors should minimize the use of the pronoun “it”.

a. Lines 86-87: Provide the reference (citation).

Author response

The reference is same Reference number 4 (now 5 in revised manuscript) and that is why the word, ‘specifically’ was used to connect both sentences. Reference number is now included.

Editor’s comment

b. Move Lines 115-118 to the first paragraph of the introduction to sharpen the introductory paragraph.

c. The introduction failed to rationalize the study using pertinent literature. Lines 109-113 should be replaced with a review of the pertient literature on TB/MH/SUD service integration. The existing literature should highlight what we already know about value, benefits, barriers, and acceptability of integrated TB/MH/SUD service delivery.

d. The last paragraph of the introduction should summarize the knowledge gaps from the existing literature including paucity of empirical literature on the study topic globally, geographical gaps (few studies in the study setting), methodological gaps (study designs, data analysis, etc), and population gap. The authors must state the specific gap the current stuy fills, the purpose of this study, and the significance of the study (who will use the evidence and how?).

Author response

In the last paragraph of the discussion, the Authors stated that this is the first time that an integrated service delivery for TB/MH/SUD was initiated in Nigeria thus acknowledging this as one remarkable strength of the study. (Suffice it to say that this observation cannot be regarded as a weakness). What the Authors did was to chronicle the series of integrated service delivery in Nigeria using that as justification for the study. In this revision, the Authors provided more points to justify the study. The Authors are of the opinion that the comments as indicated in this response responds adequately to the issues raised in the comments above.

Editor’s comment

4. Methods

a. Study setting: Why did you select the three states? Perhaps, they were the project states for the ISD for TB/MH/SUD.

b. The authors should describe the ISD for TB/MH/SUD project under the study setting.

Author response

Included. See study setting.

Editor’s comment

c. Study design, participants, and sampling: The authors should separate this sub-section into three different sub-sections to avoid missing important details of any part.

i. Lines 153-154: insert qualitative into the sentence just before study design.

ii. Participants: The subsection should appropriately titled “study population.’ The authors should describe the population from which the participants were sampled.

iii. Lines 154-159 describe a data collection procedure and should be moved to the data collection sub-section (lines 165-177).

iv. Lines 161-162: Move the sentence to the data collection sub-section.

v. Lines 175 and 177: change interview to FGD.

Author response

Corrections made

Editor’s comment

d. Data management:

i. Explain how the study derived the pre-determined themes.

ii. How many people coded the transcripts?

iii. The study seemed to have done thematic analysis but fell short of stating so. The authors should describe the process of thematic analysis.

Author response

Included.

Editor’s comment

5. Results

a. Theme 1: has three sub-themes including ISD as a worthwhile approach, positive providers’ attitude, and context of PHC.

b. Theme 2: sub-themes include advantages and barriers

i. Advantages include saves cost of transportation, reduces stress, saves time, affordable cost of care, increased awareness of the three diseases, and improved patient outcome/experiences.

ii. Barriers include mentally ill patients exposed to TB infection, lack of specialized care, increased waiting time, and TB patients risk attack from mentally ill patients.

c. Theme 3: Sub-themes include acceptance of ISD by LHWs, government approval, affordability, proximity, training of LHWs, provision of uniforms, clear roles for LHWs, employment opportunities, and informal payments to LHWs.

d. The authors must be clear about the unit of analysis. Is the unit of analysis individuals or FGDs? The usefulness of FGDs lies in the convergence or divergence of views within and across groups and states.

e. The above summary of the themes and sub-themes can be presented in a Table.

Author response

Thanks for the observation. However, the themes and sub-themes of the study are as indicated in the presentation of results.

Editor’s comment

f. The authors must be intentioanl about the choice of verbs when describing participants’ perspectives. Such verbs like applauded, pleaded, and pleased should be avoided. Use stated, mentioned, claimed, noted, suggested, and similar verbs that are emotionally neutral.

Author response

Noted with Thanks.

Editor’s comment

6. Discussion

a. The discussion should have an introductory paragraph showing the purpose of the study, and the key findings warranting further exploration across the three thematic areas.

i. Perceived value of ISD – worthwhile approach to service delivery and positive provider attitude

ii. Perceived benefits and barriers to integrated service delivery – reduced transportation cost, increased awareness, safety of mentally ill people, and lack of expertise/specialization.

iii. Acceptability of community-based ISD through lay health workers – willingness to patronize ISD by LHWs, and training/capacity-building of LHWs.

Author response

Included

Editor’s comment

b. Each paragraph should focus on one issue. However, Lines 483-500 (the fifth paragraph of the discussion) dealth with two topics – risk of mentally ill people contracting TB, and lack of medical specialization/expertise in the 3 diseases. The authors should consider separating them.

Author response

The comments on risk of mentally ill people contracting TB are contained in about four lines of the discussion which may not be able to stand as a paragraph. It may be of benefit if the two comments remain in the same paragraph since they are focused on the same concept.

Editor’s comment

c. Lines 532-540: This is not a strength of the study but the strength of the ISD project. The authors must distinguish between the study (research process and its outcome) and the ISD project. The authors should explore strengths related to knowledge gap filled.

d. Curiously, the study omitted the limitations of the study.

Author response

Limitations included

7. Conclusion

The conclusion should restate the objective of this study and succinctly summarize the policy and practice implications of the main findings of the study. Avoid ambitious or extraneous recommendations that are not based on the findings.

Author response

Conclusion revised.

Reviewer’s comment

The manuscript needs to be resubmitted with revisions

Overall evaluation

- Congratulations on the study highlighting the importance of integrated care for tuberculosis, mental health, and substance abuse, which is essential for effective tuberculosis control, especially in high-burden countries.

- Below are some comments to positively contribute to the enhancement of the study:

- Title: The title is appropriate, concise, and coherent with the study content. I recommend adding at the end of the title a colon followed by the specification of the study type, which will facilitate immediate identification of the research nature by readers.

Author report

Thanks for the kind consideration

Reviewer’s comment

- Abstract: The abstract is adequate, containing a clear synthesis aligned with the research objective.

- Introduction: The introduction covers relevant national and international concepts and data. However, the study would benefit from including a description of how tuberculosis care is provided in Nigeria, as well as access to health services in the country. In many countries, such care is offered in an integrated manner, especially driven by Primary Health Care, and international literature presents numerous promising examples that reinforce this integrated approach. It would be enriching to understand how Nigeria acts in disease control, considering that tuberculosis care still seems fragmented. Furthermore, given the study’s objective interrelating tuberculosis, mental health, and substance use as a chain perpetuating the disease, it would be pertinent to clarify how psychiatric services function in Nigeria, for example, whether they cover all types of mental disorders, including less severe cases, since it is mentioned that only hospitals fulfill this role. Although some methodological aspects briefly address this issue, we suggest that the introduction also include information about available mental health services and which substances are involved in substance abuse.

Author response

All these are well explained under study setting. Thanks.

Reviewer comment

- Methodology: It is recommended to include the following information to enrich this section:

- In the section regarding data collection instruments, highlight that the instrument used was a semi-structured interview; specify which questions were used in the focus group discussions, even if based on prior pilot studies.

- In data management, inform how many researchers participated in the transcription, and whether any software was used for this step or if it was performed manually.

- Detail the technique used for qualitative data analysis. Although the use of QDA Miner Lite v2.0.6 software was mentioned, it is important to specify the analytical technique employed, whether thematic, discourse analysis, or other, and to detail it for replicability.

Author response

Included

Results:

- I suggest removing the first paragraph of the results section and including this information

---

## [Editor Report · Decision Letter 1]

19 Nov 2025

Dear Dr. Ossai,

Thank you for submitting your manuscript to PLOS ONE. After careful consideration, we feel that it has merit but does not fully meet PLOS ONE’s publication criteria as it currently stands. Therefore, we invite you to submit a revised version of the manuscript that addresses the points raised during the review process.

We look forward to receiving your revised manuscript.

Kind regards,

Daniel Chukwuemeka Ogbuabor, Ph.D., M.D.

Academic Editor

PLOS ONE

Journal Requirements:

Additional Editor Comments (if provided):

Introduction

1. The current version of the introduction is a significant improvement over the initial submission. Nevertheless, a critical gap in the review of pertinent literature highlighting what we already know about integrating TB, mental health and substance disorders, is missing in the introduction. I have included some published articles (this is by no means exhaustive) to highlight existing studies that the authors could use to fill this important gap in the introduction. The authors should situate this review immediately after the current paragraph 3.

I. Sweetland, A. C., Gruber Mann, C., Fernandes, M. J., Silva, F. V. S. de M., Matsuzaka, C., Cavalcanti, M., Fortes, S., Kritski, A., Su, A. Y., Ambrosio, J. C., Kann, B., & Wainberg, M. L. (2024). Barriers and Facilitators to Integrating Depression Treatment Within a TB Program and Primary Care in Brazil. Health Promotion Practice, 25(6), 1032–1039. https://doi.org/10.1177/15248399231183400

II. Todowede, O., Afaq, S., Adhikary, A., Kanan, S., Shree, V., Jennings, H. M., Faisal, M. R., Nisar, Z., Khan, I., Desai, G., Huque, R., & Siddiqi, N. (2023). Barriers and facilitators to integrating depression care in tuberculosis services in South Asia: a multi-country qualitative study. BMC Health Services Research, 23(1), Article 818. https://doi.org/10.1186/s12913-023-09783-z

III. Foo, C. D., Shrestha, P., Wang, L., Du, Q., García-Basteiro, A. L., Abdullah, A. S., & Legido-Quigley, H. (2022). Integrating tuberculosis and noncommunicable diseases care in low- and middle-income countries (LMICs): A systematic review. PLoS Medicine, 19(1), e1003899. https://doi.org/10.1371/journal.pmed.1003899

IV. Afaq, S., Ayub, A., Faisal, M. R., Nisar, Z., Zala, Rehman, A. ur, Ahmed, A., Todowede, O., & Siddiqi, N. (2024). Depression care integration in tuberculosis services: A feasibility assessment in Pakistan. Health Expectations : An International Journal of Public Participation in Health Care and Health Policy, 27(1), e13985-n/a. https://doi.org/10.1111/hex.13985

V. Heunis, C., & Kigozi-Male, G. (2024). Exploring Managers’ Insights on Integrating Mental Health into Tuberculosis and HIV Care in the Free State Province, South Africa. International Journal of Environmental Research and Public Health, 21(11), 1528. https://doi.org/10.3390/ijerph21111528

VI. Lovero, K. L., Lammie, S. L., van Zyl, A., Paul, S. N., Ngwepe, P., Mootz, J. J., Carlson, C., Sweetland, A. C., Shelton, R. C., Wainberg, M. L., & Medina-Marino, A. (2019). Mixed-methods evaluation of mental healthcare integration into tuberculosis and maternal-child healthcare services of four South African districts. BMC Health Services Research, 19(1), Article 83. https://doi.org/10.1186/s12913-019-3912-9

Methods

2. In lines 178 and 195, change interview to focus group discussion.

3. In the data management sub-section, be explicit by stating that the study analyzed the data thematically before describing the process.

4. Move lines 225-229, “Interviewer characteristics” subsection to the “Study instrument and data collection method” subsection. Interviewer characteristics does not need to be a separate subsection.

Discussion

5. In line with the review of pertinent literature in the introduction, the authors should revise the discussion, by comparing the findings of the current study to results of existing scholarship.

6. In line 512, replace “a number of participants” with “few participants”.

7. Lines 512-529 contains 3 key findings that should be discussed in separate paragraphs – safety of mentally ill patients from contracting TB, a lack of medical specialization, and increased waiting time. The authors’ argument that the comments on risk of mentally ill people contracting TB are contained in about four lines of the discussion, which may not be able to stand as a paragraph, is defective. Secondly, it is true that risk of contracting TB and lack of medical specialization ‘are focused on the same concept.’ The authors should be diligent enough to discuss the findings in separate paragraphs because they can stand own their own.

8. Move lines 543-544 to the preceding paragraph (lines 531-541) to ensure smooth transition to the succeeding paragraph.
---

## [Author Response · Author response to Decision Letter 2]

22 Nov 2025

Response to Reviewers

Editor’s comment

Introduction

1. The current version of the introduction is a significant improvement over the initial submission. Nevertheless, a critical gap in the review of pertinent literature highlighting what we already know about integrating TB, mental health and substance disorders, is missing in the introduction. I have included some published articles (this is by no means exhaustive) to highlight existing studies that the authors could use to fill this important gap in the introduction. The authors should situate this review immediately after the current paragraph 3.

I. Sweetland, A. C., Gruber Mann, C., Fernandes, M. J., Silva, F. V. S. de M., Matsuzaka, C., Cavalcanti, M., Fortes, S., Kritski, A., Su, A. Y., Ambrosio, J. C., Kann, B., & Wainberg, M. L. (2024). Barriers and Facilitators to Integrating Depression Treatment Within a TB Program and Primary Care in Brazil. Health Promotion Practice, 25(6), 1032–1039. https://doi.org/10.1177/15248399231183400

II. Todowede, O., Afaq, S., Adhikary, A., Kanan, S., Shree, V., Jennings, H. M., Faisal, M. R., Nisar, Z., Khan, I., Desai, G., Huque, R., & Siddiqi, N. (2023). Barriers and facilitators to integrating depression care in tuberculosis services in South Asia: a multi-country qualitative study. BMC Health Services Research, 23(1), Article 818. https://doi.org/10.1186/s12913-023-09783-z

III. Foo, C. D., Shrestha, P., Wang, L., Du, Q., García-Basteiro, A. L., Abdullah, A. S., & Legido-Quigley, H. (2022). Integrating tuberculosis and noncommunicable diseases care in low- and middle-income countries (LMICs): A systematic review. PLoS Medicine, 19(1), e1003899. https://doi.org/10.1371/journal.pmed.1003899

IV. Afaq, S., Ayub, A., Faisal, M. R., Nisar, Z., Zala, Rehman, A. ur, Ahmed, A., Todowede, O., & Siddiqi, N. (2024). Depression care integration in tuberculosis services: A feasibility assessment in Pakistan. Health Expectations : An International Journal of Public Participation in Health Care and Health Policy, 27(1), e13985-n/a. https://doi.org/10.1111/hex.13985

V. Heunis, C., & Kigozi-Male, G. (2024). Exploring Managers’ Insights on Integrating Mental Health into Tuberculosis and HIV Care in the Free State Province, South Africa. International Journal of Environmental Research and Public Health, 21(11), 1528. https://doi.org/10.3390/ijerph21111528

VI. Lovero, K. L., Lammie, S. L., van Zyl, A., Paul, S. N., Ngwepe, P., Mootz, J. J., Carlson, C., Sweetland, A. C., Shelton, R. C., Wainberg, M. L., & Medina-Marino, A. (2019). Mixed-methods evaluation of mental healthcare integration into tuberculosis and maternal-child healthcare services of four South African districts. BMC Health Services Research, 19(1), Article 83. https://doi.org/10.1186/s12913-019-3912-9

Author response

Thanks for the observation and the referred manuscripts as indicated in the comment above. These are very relevant manuscripts when it comes to integration of services. However, a close scrutiny will reveal that the focus of the studies as listed differed so much from ours which elicited the views of beneficiaries of integrated service delivery and their caregivers. It is important to point out that almost all the studies were referenced in the discussion section where they suited the focus of our manuscript.

Editor’s comment

Methods

2. In lines 178 and 195, change interview to focus group discussion.

Author response

Corrections made. Thanks.

Editor’s comment

3. In the data management sub-section, be explicit by stating that the study analyzed the data thematically before describing the process.

Author response

Included

Editor’s comment

4. Move lines 225-229, “Interviewer characteristics” subsection to the “Study instrument and data collection method” subsection. Interviewer characteristics does not need to be a separate subsection.

Author response

This has been effected.

Editor’s comment

Discussion

5. In line with the review of pertinent literature in the introduction, the authors should revise the discussion, by comparing the findings of the current study to results of existing scholarship.

Author response

They referred manuscripts were included in the discussion section.

Editor’s comment

6. In line 512, replace “a number of participants” with “few participants”.

Author response

Correction has been made.

Editor’s comment

7. Lines 512-529 contains 3 key findings that should be discussed in separate paragraphs – safety of mentally ill patients from contracting TB, a lack of medical specialization, and increased waiting time. The authors’ argument that the comments on risk of mentally ill people contracting TB are contained in about four lines of the discussion, which may not be able to stand as a paragraph, is defective. Secondly, it is true that risk of contracting TB and lack of medical specialization ‘are focused on the same concept.’ The authors should be diligent enough to discuss the findings in separate paragraphs because they can stand own their own.

Author response

This has been done. Thanks.

Editor’s comment

8. Move lines 543-544 to the preceding paragraph (lines 531-541) to ensure smooth transition to the succeeding paragraph.

Author response

Thanks for the observation. However, moving lines 543-544 to lines 531-541 will undermine the opening statement in discussing the third and most important theme of the study which may not be appropriate. Moreover, lines 543-544 is another opening statement that supports the statements in the preceding paragraph (lines 531-541) which is the opening statement of the third theme of the study.

---

## [Decision Letter · Decision Letter 2]

3 Mar 2026

Examining community perspectives on integrated service delivery for tuberculosis, mental health and substance use disorder in Nigeria: a qualitative study

PONE-D-25-42838R2

Dear Dr. Ossai,

We’re pleased to inform you that your manuscript has been judged scientifically suitable for publication and will be formally accepted for publication once it meets all outstanding technical requirements.

Kind regards,

Farrukh Ishaque Saah, MPhil Population and Health

Guest Editor

PLOS One

Additional Editor Comments (optional):

Reviewers' comments:

Reviewer's Responses to Questions

**Comments to the Author**

Reviewer #1: All comments have been addressed

Reviewer #3: All comments have been addressed

2. Is the manuscript technically sound, and do the data support the conclusions?

Reviewer #1: Yes

Reviewer #3: Yes

3. Has the statistical analysis been performed appropriately and rigorously?

Reviewer #1: N/A

Reviewer #3: N/A

4. Have the authors made all data underlying the findings in their manuscript fully available?

Reviewer #1: Yes

Reviewer #3: Yes

5. Is the manuscript presented in an intelligible fashion and written in standard English?

Reviewer #1: Yes

Reviewer #3: Yes

Reviewer #1: (No Response)

Reviewer #3: The authors have adequately addressed all comments raised in the previous round of review. The manuscript is technically sound and employs appropriate qualitative methodology, including well-described focus group discussions, purposive sampling, thematic analysis, and evidence of data saturation. The findings are clearly presented and support the conclusions drawn.

The manuscript is well organised and written in clear, standard English. I have no concerns regarding research ethics, publication ethics, or dual publication. Overall, the manuscript is suitable for publication.

.

Reviewer #1: **Yes:** Rosiane Davina da SilvaRosiane Davina da SilvaRosiane Davina da SilvaRosiane Davina da Silva

Reviewer #3: **Yes:** Grace KakaireGrace KakaireGrace KakaireGrace Kakaire

---

## [Editor Report · Acceptance letter]

PONE-D-25-42838R2

PLOS One

Dear Dr. Ossai,

I'm pleased to inform you that your manuscript has been deemed suitable for publication in PLOS One. Congratulations! Your manuscript is now being handed over to our production team.

Kind regards,

on behalf of

Dr. Farrukh Ishaque Saah

Guest Editor

PLOS One